# Demographic patterns of human antibody levels to *Simulium damnosum* s.l. saliva in onchocerciasis-endemic areas: An indicator of exposure to vector bites

**Laura Willen** [1,2]*, **Philip Milton** [3], **Jonathan I. D. Hamley** [3], **Martin Walker** [4], **Mike Y. Osei-Atweneboana** [5], **Petr Volf** [1], **Maria-Gloria Basáñez** [3], **Orin Courtenay** [6]*

1 Department of Parasitology, Faculty of Science, Charles University, Prague, Czech Republic, 2 Centre for the Evaluation of Vaccinations, Vaccine and Infectious Disease Institute, University of Antwerp, Wilrijk, Belgium, 3 MRC Centre for Global Infectious Disease Analysis and London Centre for Neglected Tropical Disease Research, Department of Infectious Disease Epidemiology, School of Public Health, Imperial College London, London, United Kingdom, 4 London Centre for Neglected Tropical Disease Research and Department of Pathobiology and Population Sciences, Royal Veterinary College, Hatfield, United Kingdom, 5 Biomedical and Public Health Research Unit, CSIR-Water Research Institute, Accra, Ghana, 6 Zeeman Institute for Systems Biology & Infectious Disease Epidemiology Research and School of Life Sciences, University of Warwick, Coventry, United Kingdom

* laura.willen@gmail.com (LW); orin.courtenay@warwick.ac.uk (OC)

**Data Availability Statement:** The summarized data supporting the conclusions of this article are included within the manuscript and/or Supporting

## Abstract

### Background

In onchocerciasis endemic areas in Africa, heterogenous biting rates by blackfly vectors on humans are assumed to partially explain age- and sex-dependent infection patterns with *Onchocerca volvulus*. To underpin these assumptions and further improve predictions made by onchocerciasis transmission models, demographic patterns in antibody responses to salivary antigens of *Simulium damnosum* s.l. are evaluated as a measure of blackfly exposure.

### Methodology/Principal findings

Recently developed IgG and IgM anti-saliva immunoassays for *S. damnosum* s.l. were applied to blood samples collected from residents in four onchocerciasis endemic villages in Ghana. Demographic patterns in antibody levels according to village, sex and age were explored by fitting generalized linear models. Antibody levels varied between villages but showed consistent patterns with age and sex. Both IgG and IgM responses declined with increasing age. IgG responses were generally lower in males than in females and exhibited a steeper decline in adult males than in adult females. No sex-specific difference was observed in IgM responses.

### Conclusions/Significance

The decline in age-specific antibody patterns suggested development of immunotolerance or desensitization to blackfly saliva antigen in response to persistent exposure. The variation

Information files. The raw data are available on reasonable request to be used within the context of this study and ethical approvals at the University of Warwick data repository (URL: http://wrap. warwick.ac.uk/160693/).

**Funding:** The work was supported by the Global Challenges Research Fund (Networks in Vector Borne Disease Research Gnatwork) BBSRC grant (BB/R005362/1) to OC, MGB, PV and MYO-A. LW and PV were further supported by the Ministry of Education of the Czech Republic through the European Regional Development Fund (project CePaViP, CZ.02.1.01/0.0/0.0/16_019/0000759). OC acknowledges the continued support of the Wellcome Trust (https://wellcome.ac.uk) Strategic Translation Award (WT091689MF). MGB, JIHD and PM acknowledge funding from the Medical Research Council (MRC) Centre for Global Infectious Disease Analysis (grant no. MR/ R015600/1), jointly funded by the UK MRC and the UK Foreign, Commonwealth & Development Office (FCDO), under the MRC/FCDO Concordat agreement and is also part of the European and Developing Countries Clinical Trials Partnership (EDCTP2) programme supported by the European Union. MGB, JIHD and MW acknowledge funding from the NTD Modelling Consortium (https://www. ntdmodelling.org, [grant no. OPP1184344] by the Bill & Melinda Gates Foundation (https://www. gatesfoundation.org/). PM is supported by a UK Medical Research Council doctoral training award. The funders had no role in study design, data collection and analysis, decision to publish, or preparation of the manuscript.

**Competing interests:** The authors have declared that no competing interests exist.

between sexes, and between adults and youngsters may reflect differences in behaviour influencing cumulative exposure. These measures of antibody acquisition and decay could be incorporated into onchocerciasis transmission models towards informing onchocerciasis control, elimination, and surveillance.

## Author summary

Onchocerciasis, a disease caused by the helminth parasite *Onchocerca volvulus*, is transmitted by the bites of female *Simulium* blackflies. The disease is still endemic in many African countries, and the World Health Organization has proposed elimination of its transmission in 12 countries by 2030. Understanding the heterogeneity in human exposure to vector bites can help discern which portion of the population is at higher risk of acquiring/ transmitting infection and is fundamental to identifying target groups for serological monitoring and transmission control. Traditionally, blackfly biting rates are estimated by performing human landing catches, a method that is often considered unethical and which can be unreliable as a representative measure. Therefore, we used our recently developed immunoassays to measure human antibody responses to antigens contained in the saliva of blackflies and deposited into human skin when they bloodfeed. In onchocerciasis endemic communities in Ghana, we measured antibody responses to understand age- and/or sex-related demographic patterns in vector exposure. We observed lower antibody responses in males compared to in females, and a substantial decline with increasing age, suggesting that high blackfly biting pressure induces desensitization in the human host.

## Introduction

Female blackflies of the *Simulium damnosum* sensu lato (s.l.) complex are the predominant vectors of *Onchocerca volvulus* in Africa. This filarial nematode causes human onchocerciasis, commonly known as river blindness and proposed for elimination of transmission (EOT) in 12 countries by 2030 [1]. Repeat exposure to bites of infective blackflies is a key driver of parasite acquisition, and high biting rates are important determinants of transmission intensity and resurgence following control interventions, particularly those based on mass drug administration (MDA) of ivermectin [2,3]. Therefore, 'stop-MDA surveys' and post-treatment surveillance protocols would be improved by monitoring exposure to vector bites in addition to exposure to the parasite and assessment of residual infection in informative age and sex groups depending on the epidemiological setting [4,5]. Annual biting rates (ABR, the number of bites/person/year) are predominantly estimated by performing human landing catches (HLCs) [6], but this is both labour-intensive and often considered unethical. Furthermore, HLC methods likely lead to biased estimates of ABRs as they are typically performed on a few adult males at the riverbank close to blackfly breeding sites, where biting rates are likely to be higher and vector collectors are maximally exposed [7]. Thus, HLCs may not capture the true exposure representative of the community, age or sex groups or their activity patterns. Furthermore, when blackfly densities are high, HLCs may be unable to capture all biting events thereby underestimating the biting rate [8,9]. Applying novel methods to measure individual and population exposure to blackfly vector bites would greatly inform our understanding of

exposure patterns to onchocerciasis and inform mathematical transmission modelling of control and elimination [2].

Proteins in the saliva of blood-seeking arthropod vectors of human and veterinary diseases provoke an immunomodulatory response in vertebrates following their exposure to vector bites (e.g. by mosquitoes, sand flies, triatomine bugs, and tsetse flies) [10]. Measurement of host anti-saliva antibodies has proven a useful surrogate marker to monitor individual host biting rates [11–15], seasonal variations in exposure and infection [16,17], and to evaluate vector control interventions [18–22]. Immunoassays to measure human anti-saliva antibody responses to blackfly bites have not been available until recently developed by the authors, specifically to measure human IgG and IgM antibody responses to saliva of *S. damnosum* s.l. which is the main vector in Africa including the study sites in Ghana [23].

In the present study we report on the demographic patterns revealed by applying these assays to community residents in the Bono East region of Ghana, a savannah setting. This area was under vector control during the Onchocerciasis Control Programme in West Africa (OCP, 1974–2002) [24,25], and is where the first ivermectin community trials were conducted in the late 1980's [26]. However, transmission persists despite many years of annual MDA, even after adopting biannual MDA in 2010 [27,28]. With the risk of residual transmission leading to possible resurgence of human infections if MDA were to stop [3], the area has been the subject of comprehensive entomological [27,29–31], parasitological [32] and parasite genomic [33] research, and a source of valuable epidemiological and entomological data for parameterizing onchocerciasis transmission models [34–36]. Due to the lack of reliable tools to independently measure human exposure to vector bites, mathematical models have assumed age- and sex-dependent exposure rates based on patterns of human infection with *O. volvulus*. However, modelling outputs used to inform EOT and surveillance strategies are sensitive to such patterns [4,37]. The aim of this study is to apply our novel human IgG and IgM immunoassays as indicators of individual biting exposure, both between and within (by age and sex) onchocerciasis-endemic communities. The generated demographic patterns will help to scrutinize mathematical modelling assumptions.

## Material and methods

### Ethical statement

Ethical clearance was obtained from the Council for Scientific and Industrial Research (CSIR) Institutional Review Board (RPN008/CSIR-IRB/2019) in Accra, Ghana. Residents were informed of the objectives of the study and participants provided fully informed written consent; parents or guardians provided consent for <18-year-olds. At the time of the study, ivermectin treatment against onchocerciasis was administered as part of the on-going national biannual MDA programme [38]. Each participant was rewarded with a bar of washing soap, a commercial sachet of malt drink powder (Milo, Nestlé) and a can of condensed milk. Blackfly collections were carried out by local OCP vector collectors following standard HLC techniques [39]. Individual identifiers on data records were anonymized prior to analyses and data storage.

### Study area

The study was conducted during the wet season in August 2019 in rural villages in an onchocerciasis-endemic area of the Pru river basin, Pru District, in the Bono East region of Ghana. Fig 1 provides a map of the study area and the location of the study communities, created using the 'ggplot2' and the 'rworldmap' package in R software [40,41]. The shapefiles for the base layers of the map were retrieved from the Database of Global Administrative Areas

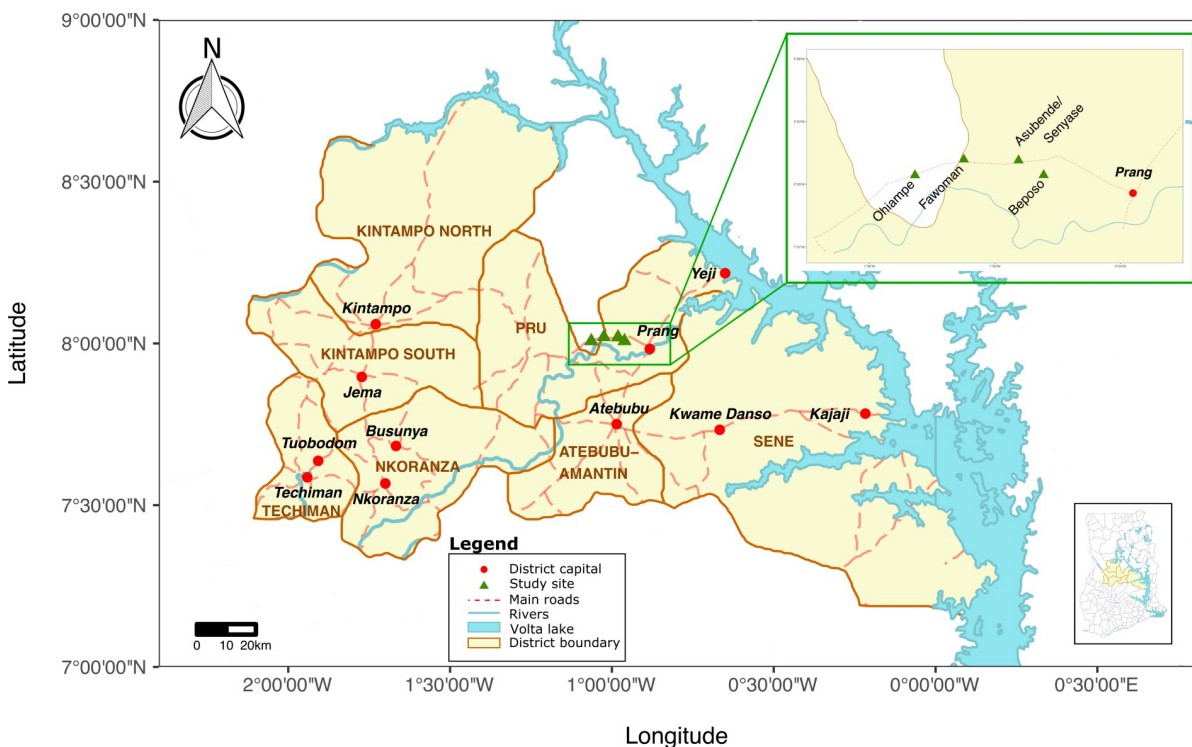

**Fig 1. Map of Ghana indicating study communities in the Pru District, Bono East region.** Shapefiles for the base layers of the map retrieved from GADM (https://gadm.org/download_country.html) and DIVA-GIS (http://www.diva-gis.org) under a CC BY license, with permission from GADM and DIVA-GIS.

(GADM, https://gadm.org/download_country.html) and from DIVA-GIS (http://www.diva-gis.org) under a CC BY license, with permission from GADM and DIVA-GIS.

### Recruitment, blood sampling and demographic information

Villages were selected on the criteria that (a) there was evidence of exposure to *S. damnosum* s.l. vector bites [30,42], (b) they were under current ivermectin treatment as part of the national onchocerciasis control programme [43], (c) the villages comprised >200 residents, and (d) the villages represented variable ABRs based on previous records [30,42]. Five villages were selected; due to the proximity of two of the villages, these were grouped into a single cluster (Asubende and Senyase (ASU/SEN) resulting in four village clusters. These included ASU/SEN, with high biting rates (400–850 flies/person/month), Beposo [BEP] with moderate values (100–350), and Fawoman [FAW] and Ohiampe [OHI] with lower biting rates (25–175 flies/person/month) [30,42].

To obtain blood samples from the village residents, two recruitment approaches were adopted: in villages with <300 residents (ASU/SEN, FAW, and OHI) all residents were invited to participate. In the larger village (BEP) an age/sex stratified sample was identified based on assigning random numbers generated using R software [44], to the pages of the paper-based census records compiled by the MDA programme. The households contained on randomly selected pages were noted and invited to participate until reaching an estimated sample of houses containing 250 individuals as required from statistical calculations (i.e. ~1000 individuals in total across the four village clusters). Children younger than 4 years old were not recruited.

**Table 1. Study communities, their geographical coordinates, elevation and numbers of people sampled and tested for IgG and IgM per community in the Pru District, Ghana.**

| Community/Cluster | Coordinates (degrees, minutes, seconds) | | Elevation (masl) | No. sampled | |
| --- | --- | --- | --- | --- | --- |
| | | | | Tested for each immunoglobulin (%) | |
| | Long | Lat | | IgG | IgM |
| **Asubende/Senyase [ASU/SEN]** | 08˚01'08.8"N | 00˚58'52.4"W | 153.3 | 186 | 97 |
| | | | | 100% | (52.2%) |
| **Beposo [BEP]** | 08˚00'26.7"N | 000˚57'40.2"W | 118.0 | 253 | 139 |
| | | | | 100% | (54.9%) |
| **Fawoman [FAW]** | 08˚01'11.4"N | 001˚01'29.3"W | 102.4 | 263 | 124 |
| | | | | 100% | (47.2%) |
| **Ohiampe [OHI]** | 08˚00'26.2"N | 001˚03'49.5"W | 114.3 | 256 | 140 |
| | | | | 100% | (54.7%) |
| **Total** | | | | 958 | 500 |
| | | | | 100% | (52.2%) |

Long: Longitude; Lat: Latitude; masl: metres above sea-level

A total of 958 participants were finally recruited, all tested for IgG antibodies, and a subset of 500 individuals tested for IgM antibodies (Table 1). For age stratification purposes, the identified populations in all villages were divided into eight age categories (in years): 5–10, 11–20, 21–30, 31–40, 41–50, 51–60, 61–70, > 71, aiming to recruit 32 residents per age class in each village, split equally between the sexes. In the case of IgM, samples were selected by random number generation assigned to the ordered age-stratified full list of samples and aimed to test around 40–60% of the samples tested by IgG per age class, with some variability in percentage tested in the oldest age group. The numbers tested per age and sex strata per village cluster are shown in Figs A and B and Table C in S1 File. The geographical coordinates, elevation and number of individuals sampled and tested for each immunoglobulin per village cluster are shown in Table 1.

Participants in each village/cluster were invited to the local school and assigned a subject/sample identification number written on a personalized card. Two to four ml of blood were collected into Ethylene Diamine Tetra Acetic acid (EDTA) tubes by venipuncture and kept cool for 2–3 hours in an insulated cool box until centrifugation at 2500 rpm for 15 min to separate the plasma. Samples were stored at 7˚ C until ELISA testing. Metadata on name, age, sex, number of years of residence, house number, name of household head, and history of clinical onchocerciasis were collected.

## Immunoassays to measure human exposure to blackfly bites

Enzyme-linked immunosorbent assays (ELISA) previously developed were performed to measure anti-*S. damnosum* s.l. saliva IgG and IgM human responses [23]. Briefly, host-seeking *S. damnosum* s.l. females were collected following standard OCP vector collector techniques in one study location (ASU), near the Pru river [39]. All collected flies were stored in a cool box until dissected on the same day. The collected flies were anesthetized in a –20˚ C freezer for 10 min after which their salivary glands were removed, and aliquots stored in Tris-buffered saline (TBS) (one gland per µl TBS, pH 7.5) at –20˚ C until further use. Flat-bottom 96-well microtiter plates (ThermoFisher Scientific) were coated with blackfly salivary gland

homogenate (SGH) with 0.2 µg SGH/well (for IgG ELISA) or 0.025 µg SGH/well (for IgM ELISA) and incubated overnight at 4˚ C. The plates were blocked with 6% non-fat dried milk (Bio-Rad) in phosphate-buffered saline (PBS) with 0.05% Tween 20 (PBS-Tw) and incubated with plasma diluted 1/100 (IgG ELISA) or 1/50 (IgM ELISA) in 2% non-fat dried milk. The plates were washed and incubated with peroxidase-conjugated anti-human IgG (1/1,000) or IgM antibody (1/70,000) (Sigma-Aldrich; Bethyl Laboratories, Inc). The ELISA was developed using an orthophenylendiamine (OPD) solution in a phosphate-citrate buffer (pH 5.5) with 0.1% hydrogen peroxide. The reaction was stopped after 5 min with 10% sulfuric acid and the absorbance, optical density (OD) value was measured at 492 nm using a Tecan Infinite M200 microplate reader (Schoeller). Further details of blackfly collection, dissection, and the immunoassays were previously reported [23].

## Data standardization

All plasma samples were tested in duplicate. Samples with a coefficient of variation (CoV) of more than 20% were retested. Each plate included a blank control, of which the OD value was subtracted from the sample OD values. A set of two positive (PC) and two negative control (NC) samples were included in each plate to correct for inter-plate variability according to the following formula: Standardized Optical Density (SOD) = $OD_{sample}$/(average $OD_{PC}$−average $OD_{NC}$). Furthermore, a PC sample was titrated in duplicate on three separate plates at seven serial dilutions from 1/50 to 1/3,200. The average of the three log-logit transformed standard curves was used to convert and standardize sample SOD values which are reported below as anti-saliva antibody arbitrary units/ml.

## Sample size calculation

Sample sizes were calculated to achieve 90% statistical power (with type I error, $\alpha = 0.05$) to detect a difference in mean IgG SOD antibody levels between males and females, for an unpaired two-sample effect size $D_{Cohen}$ of 0.23 and a variance in SOD of 0.16 [45]. By adding 15% to the calculated sample size as a correction factor for subsequent non-parametric statistical testing, 460 people per sex group were required [46]. This estimate is also based on the reasonable assumption that the differences in the mean antibody responses of the sexes between clusters would be minimal as confirmed by subsequent analyses (test of village cluster × sex interaction term: $P>0.243$). The equivalent statistical power to detect a difference in IgM responses between sexes with an effect size $D_{Cohen}$ of 0.3 and a variance in SOD of 0.16 was 90%. Calculations were made in the R package 'power' using a two-sided t-test.

## Statistical analyses

Differences between clusters or sex categories were statistically tested by Wilcoxon rank sum or Kruskal-Wallis tests with post-hoc Holm adjustment for multiple comparisons. Changes in immune responses with age and/or sex were tested by fitting generalized linear models (GLM) to the IgG and IgM anti-blackfly saliva antibody values, where a gamma distribution and log-link function gave the best fit by log-likelihood goodness-of-fit statistics. Age, sex and cluster interaction terms were tested, treating age as a continuous variable. Residence duration (years) and age showed high multicollinearity; hence only age was retained in the models. Correlation coefficients ($r_s$) between median IgG and IgM responses by age and sex were estimated by the Spearman's rank method. All statistical analyses were performed in R software [44], and graphical representations created using the 'ggplot2' package in R [40].

## Results

### Univariate analyses of population antibody response distributions

A total of 958 participants (186–263 per village cluster) between the ages of ≥ 5 to < 96 years old were recruited and sampled for anti-saliva IgG antibodies; demographic characteristics are summarized in Table 2 and in Figs A and B in S1 File. No differences were observed in the age compositions between village clusters or between sexes, nor in the frequencies of participants per age group (Figs A and B in S1 File). Thus, it is unlikely that the variations in age-sex compositions of village clusters were a source of statistical bias.

Univariate analyses of demographic variables detected significant differences between the median IgG responses among village clusters ($P < 0.001$) (Fig 2A), and the median IgG responses of males and females ($P < 0.05$) (Fig 2B). Equivalent differences were not observed in IgM responses (Fig 3); however, both IgG and IgM median antibody responses declined with age ($P < 0.001$) (Figs 2D and 3D). A breakdown of the IgG and IgM antibody distribution with age per individual cluster is visualized in Figs D and E in S1 File.

### Multivariate analyses of IgG responses according to village cluster, sex and age

Accounting for cluster ID, sex, age, and the age x sex interaction term, multivariate models revealed that IgG responses were significantly lower in BEP than in ASU/SEN ($P < 0.01$), similar between ASU/SEN and FAW ($P = 0.406$), and higher in OHI than in ASU/SEN ($P < 0.001$) (Figs 4A and 5). Generally, males exhibited lower IgG responses than females ($P < 0.001$) (Figs 4B and 5). Fig 5 illustrates the (exponentiated) regression coefficient estimates of the final model (summarized in Table F in S1 File).

**Table 2. Summary statistics of participants and median antibody titres by immunoglobulin and village/cluster.**

| Assay | Village | Participants (n) | Median age (Q1 –Q3) (in years) | Sex ratio (M:F) | Median duration of residency (Q1 –Q3) (in years) | Median antibody titre (units/ml) | Reported biting rate |
|---|---|---|---|---|---|---|---|
| IgG | All | 958 | 21.0 (11.0–42.0) | 0.84:1 (437:521) | 16.0 (8.0–32.0) | 2478.13 (1550.05–3413.36) | NA |
| | ASU/SEN | 186 | 24.0 (14.0–39.0) | 1:1 (93:93) | 20.0 (10.0–35.0) | 2507.34 (1699.94–3125.95) | High |
| | BEP | 253 | 25.0 (11.0–46.0) | 0.83:1 (115:138) | 24.5 (11.0–46.0) | 1951.63 (1269.37–2822.26) | Moderate |
| | FAW | 263 | 22.0 (10.0–40.0) | 0.71:1 (109:154) | 13.0 (6.0–26.0) | 2590.61 (1662.54–3535.01) | Low |
| | OHI | 256 | 17.0 (11.0–40.5) | 0.88:1 (120:136) | 14.0 (7.0–24.0) | 2956.20 (1703.47–4430.85) | Low |
| IgM | All | 500 | 21.5 (10.0–42.0) | 0.85:1 (230:270) | 17.0 (7.0–35.0) | 3628.43 (1687.59–9923.29) | NA |
| | ASU/SEN | 97 | 23.0 (12.25–36.75) | 1.37:1 (56:41) | 21.0 (9.25–35.75) | 2956.66 (1496.31–7690.82) | High |
| | BEP | 137 | 24.0 (10.0–45.0) | 0.72:1 (58:81) | 24.0 (10.0–45.0) | 4021.58 (1462.01–10396.13) | Moderate |
| | FAW | 124 | 23.0 (9.0–40.0) | 0.77:1 (54:70) | 11.0 (6.0–29.0) | 4102.16 (2148.38–10473.40) | Low |
| | OHI | 140 | 17.5 (9.0–42.0) | 0.79:1 (62:78) | 15.0 (7.0–25.0) | 3569.30 (1561.18–10754.69) | Low |

The interquartile range (Q1 and Q3) of age, duration of residency, and the antibody titres are shown in brackets. Reported biting rates are according to [30,42]. All: All clusters together; village clusters: ASU/SEN: Asubende/Senyase; BEP: Beposo; FAW: Fawoman; OHI: Ohiampe. n: number; M: males; F: females; NA: not applicable

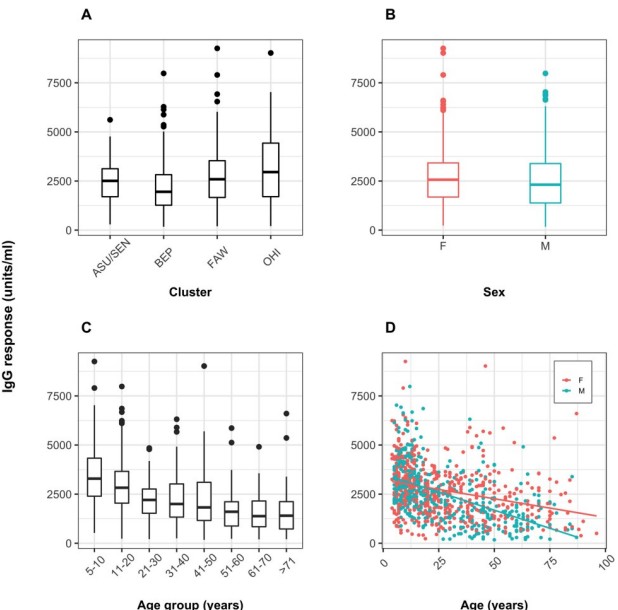

**Fig 2. Human anti-blackfly saliva IgG antibody responses according to cluster, sex, and age.** Boxplots showing the distributions of IgG responses (A) by cluster, (B) by sex, (C) by age, and scatterplot (D) by age and sex (points) showing the best-fit lines (solid lines). IgG levels are shown as units/ml. F: female; M: male. Village clusters: ASU/SEN: Asubende/ Senyase; BEP: Beposo; FAW: Fawoman; OHI: Ohiampe.

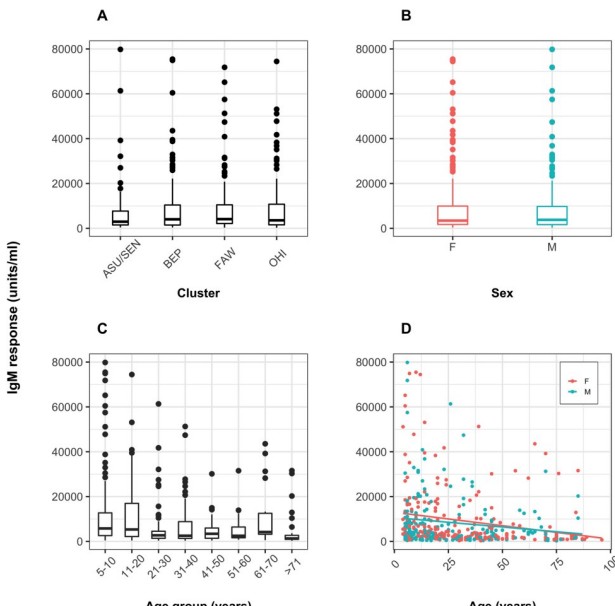

**Fig 3. Human anti-blackfly saliva IgM antibody responses according to cluster, sex and age.** Boxplots showing the distributions of IgM responses (A) by cluster, (B) by sex, (C) by age, and scatterplot (D) by age and sex (points) showing the best-fit lines (solid lines). IgM levels are shown as units/ml. F: female; M: male. Village clusters: ASU/SEN: Asubende/ Senyase; BEP: Beposo; FAW: Fawoman; OHI: Ohiampe.

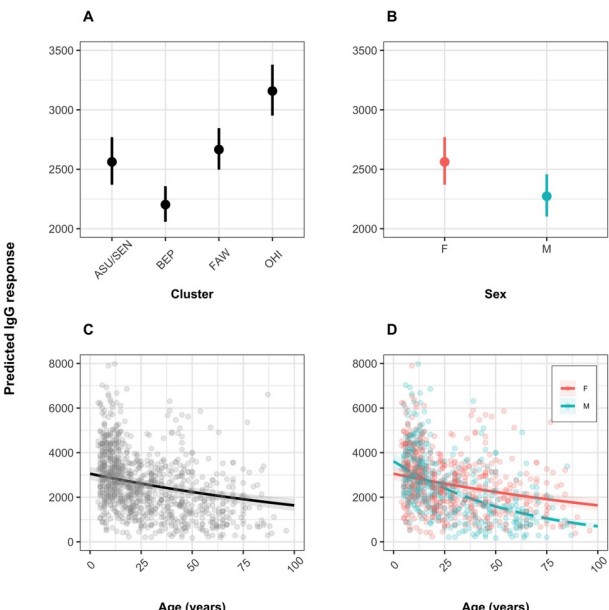

**Fig 4. Regression effect plots for all explanatory variables contained in the final model for the IgG responses.** Each plot visualizes the effect of a specific explanatory variable while fixing the others to their reference level. The panels show the variations in responses between village clusters (A), sex (B), age (C), and an interaction between age and sex (D). The vertical lines in panel A and B and the shaded regions around the lines in panel C and D represent the 95% confidence intervals. M = Male; F = Female. Village clusters: ASU/SEN: Asubende/Senyase; BEP: Beposo; FAW: Fawoman; OHI: Ohiampe.

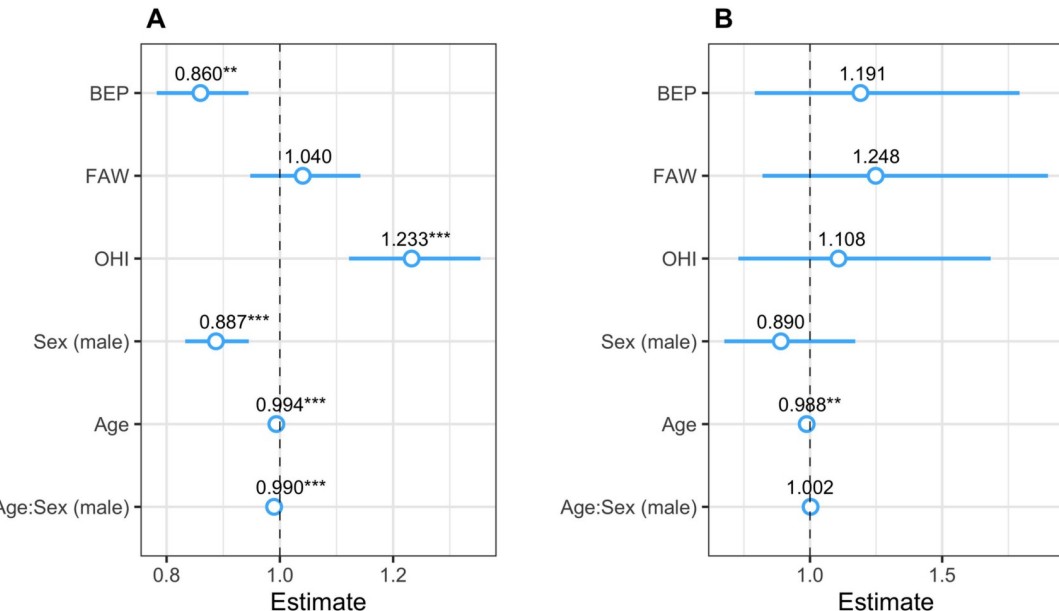

**Fig 5. Exponentiated regression coefficient estimates for the final model per assay.** The regression coefficient estimates for the final model are shown for IgG responses (A), and IgM responses (B). The circles represent the exponentiated mean coefficient estimate, the value shown above the corresponding circle, and the horizontal lines represent the 95% confidence intervals of the estimate. Significance levels *** $P<0.001$; ** $P<0.01$; * $P<0.05$. The referents were Asubende/Senyase for village clusters, and females for both sex and the age × sex interaction term. Village clusters: BEP: Beposo; FAW: Fawoman, OHI: Ohiampe.

IgG anti-saliva antibody responses decreased with increasing age in both sexes ($P<0.001$) (Figs 4C and 5), but the response in males declined at a faster rate than in females (test of age × sex interaction term; $P<0.001$) (Figs 4D and 5). Both adult male and female participants ≥18 years old exhibited lower IgG antibody levels than those aged <18 years ($P_{females}<0.001$; $P_{males}<0.001$) (Fig 5). Males and females in pre-teenage years showed a similar antibody decline with increasing age (age × sex interaction: $P = 0.328$), whereas in adults the decline was more rapid in males than in females (age × sex interaction: $P<0.01$) (Fig 4D).

## Multivariate analyses of IgM responses according to village cluster, sex or age

Equivalent multivariate analyses of anti-saliva IgM antibody responses for 500 recruits (Table 1) indicated no significant variation between study clusters ($P_{BEP} = 0.403$; $P_{FAW} = 0.301$; $P_{OHI} = 0.632$) or sex ($P = 0.405$) (Figs 5, 6A and 6B). However, there was a general decline in response magnitude with increasing age ($P<0.01$) (Figs 5 and 6C), but in contrast to IgG, the decline with age was not significantly different between sexes (test of age × sex interaction term: $P = 0.726$) (Fig 6D). Fig 5 illustrates the (exponentiated) regression coefficient estimates of the final model (summarized in Table F in S1 File).

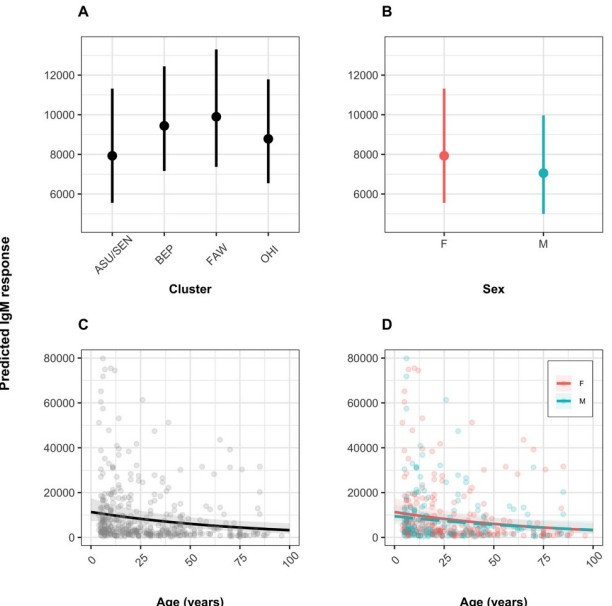

**Fig 6. Regression effect plots for all explanatory variables contained in the final model for the IgM responses.** Each plot visualizes the effect of a specific explanatory variable while fixing the others. The panels show the variations in responses between village clusters (A), sex (B), age (C), and an interaction between age and sex (D). The vertical lines in panel A and B and the shaded regions around the lines in panel C and D represent the 95% confidence intervals. M = Male; F = Female. Village clusters: ASU/SEN: Asubende/Senyase; BEP: Beposo; FAW: Fawoman; OHI: Ohiampe.

## Correlation between IgG and IgM responses

The correlation between IgG and IgM antibody responses was generally low ($r_s = 0.10$, $P<0.05$), and not dissimilar considering females or males alone (females: $r_s = 0.12$, $P<0.1$; males: $r_s = 0.08$, $P = 0.21$) (Fig F in S1 File).

## Discussion

This study measured human antibody responses against blackfly saliva as an indicator of individual bite exposure; an approach that has been validated for several vector–host systems with the exception of blackflies [47,48]. Simuliids are important disease vectors of human and bovine onchocerciasis [49], mansonelliasis caused by *Mansonella ozzardi* [50], and vesicular stomatitis virus [51], yet they have not received as much attention as sand fly, mosquito or tick vectors. Only recently, we successfully developed novel immunoassays against blackfly saliva and this is the first study to apply these tools to an epidemiological setting. The appropriate use of these assays will also improve our understanding of *O. volvulus* transmission dynamics and will be used to monitor changes in biting rates and the success of vector control interventions [23]. Heterogenous exposure to blackfly bites is an important determinant of the observed age- and sex-dependent profiles of *O. volvulus* infection [5]. Evaluating age- and sex-associated exposure patterns using empirically collected data can help inform transmission dynamics models, as these currently use assumed age- and sex-dependent exposure patterns that have, at best, been derived from fitting the models to age- and sex-specific infection profiles [5]. Such assumptions influence the choice of informative age groups for serological monitoring in stop-MDA surveys [4], and have implications for understanding the contribution of currently untreated groups to transmission and morbidity [37], as well as for the design and evaluation of potential prophylactic strategies [52]. Testing such assumptions with independently collected data is crucial for better parameterization of transmission models and improved design of epidemiological surveys and surveillance strategies. Therefore, we explored age-and sex-related patterns of IgG and IgM antibody titres against *S. damnosum* s.l. saliva across well characterized endemic communities.

Both IgG and IgM antibody levels were high in children and gradually declined with increasing age. Similar trends have been observed in human antibody responses to the saliva of several mosquito species [53–58], posing the question of whether these patterns indicate decreasing exposure to vector bites with age, or increasing immune tolerance and desensitization with persistent or cumulative saliva exposure [59–62]. Investigation of human IgG responses to sand fly saliva supports the latter proposition, with higher anti-saliva antibody responses observed in new compared to long-term residents of a sand fly-endemic region [63]. Similarly, desensitization to salivary antigens was detected in an area colonized for more than 25 years by *Aedes* mosquitoes, compared to an area where individuals had been exposed for no longer than 5 years [54]. Following that pattern, median IgG antibody responses also tended to be lower in the current study villages where higher *S. damnosum* s.l. biting rates were previously reported [30,42]. Antibody responses to *Anopheles* mosquito bites measured after the summer season of high vector abundance were considerably higher than those before the summer season; notwithstanding they consistently declined with increasing host age [54,55]. Interestingly, in that study the decline appeared to be antigen-dependent, as the trend was not detected in antibody responses to a specific recombinantly-expressed protein as opposed to the whole salivary gland homogenate [64–66]. Several immunogenic proteins were recently detected in *S. damnosum* s.l. saliva of which most were well-known salivary antigens [23]. Future expression of these in recombinant forms may be instructive and increase assay sensitivity. This is especially interesting for IgM as these antibody responses were shown to be less specific than the IgG responses [23].

The median IgG response was lower in males than females, which can be attributed to the greater rate of decline of the IgG response with increasing age in males. This may reflect sex differences in behaviour such as daily habits, occupation, education, or clothing, that influence physical exposure to blood-seeking blackflies. Male occupants of most ages are responsible for

agriculture, farming and fishing in the Pru region and may be less well covered by protective clothing, whereas women are more covered, spend more time at home performing domestic duties and/or engage in long-distance trading activities [67,68]. The current study was limited in not recording the daily activities of the participants, though biting blackflies appeared to be ubiquitous throughout the day within villages. That females show greater levels of non-specific innate and adaptive immune responsiveness than males, particularly post-puberty, suggests that hormonal involvement (reviewed in [69]) may contribute to the sex differences observed in this study.

A non-mutually exclusive alternative driving factor behind the observed decline in antibody levels may also be a lower exposure to blackfly bites with increasing age. Such patterns of exposure were predicted by fitting age- and sex-structured onchocerciasis transmission mathematical models to age- and sex-specific profiles of *O. volvulus* skin microfilariae in another African savannah setting [5]. However, the model fits also indicated that vector exposure of females should increase (rather than decrease) with age. If this is correct, and if women indeed are more intensely exposed as they age, it further supports the case for desensitization with increased long-term biting exposure. Therefore, one next step to better understand exposure patterns with age and sex is to fit dynamic models of antibody acquisition and decay to the (IgG) data obtained here [70,71]. Future studies of the molecular and cellular mechanisms that underly immune tolerance and progressive desensitization to blackfly saliva would also be most informative. Variations in IgG subclass responsiveness are also possible as shown against *Anopheles* or *Aedes* saliva [55,72]. Interestingly, human IgG4 amongst bee-keepers was found to be associated with immunotolerance to bee venom [73,74].

Less clear in this study were the age and sex-related trends in IgM responses. Although they also declined with age supporting the immunological desensitization hypothesis, IgM responses are generally shorter lived than IgG responses, hence, likely to be more indicative of recent exposure. A shorter half-life together with a lack of cumulative increase after repeat exposure may partially explain the large number of low IgM responses observed in the village residents, and the lack of correlations between individual host IgG and IgM responses, particularly as the current study was limited to cross-sectional sampling during the high biting season.

Future studies would benefit from quantifying short- and long-term kinetics in individual anti-saliva Ig responses in the context of seasonal fluctuations in vector abundance and distance to vector breeding habitats, to refine our understanding of the link between biting rates and the Ig responses [16,53,54]. In fact, it would be very informative if antibody data generated using our immunoassays could be used in spatial analyses to better understand patterns of vector–human contact with increasing distance from breeding sites. At present, mapping exercises such as those used for Onchocerciasis Elimination Mapping (OEM) collect information on the distribution of breeding sites and vector presence. Our novel tool could complement OEM to identify high-risk locations where exposure to vector bites would provide additional information to seroprevalence surveys to guide start-MDA decisions and identify informative age/sex groups for sampling [75]. Most of the information about the relationship between vector density and distance from breeding sites pertains to African savannah settings (such as those explored here), with less data available to characterize such a relationship in forest and forest-savannah mosaic settings. Therefore, if our anti-vector saliva assays could be combined with (seroprevalence) parasite exposure assays for a range of epidemiological settings, it would be possible to obtain valuable information to help elimination efforts. However, this necessitates the testing and validation of our assays for other species/cytoforms of the *S. damnosum* complex.

The predominant vector species in the Bono East region are the savannah members of the complex, *S. damnosum* sensu stricto/*S. sirbanum* [27,30]. It remains to be established if the

anti-*S. damnosum* s.l. IgG and IgM antibody responses represent a *damnosum* complex-specific marker, or if member-specific markers of exposure would be more sensitive. We acknowledge that more data need to be collected to ascertain the validity of our immunoassays to reliably measure exposure to vector bites. Regarding empirical approaches, there are no experimental or observational data yet to precisely quantify the relationship between anti-salivary antibody levels and the number of vector bites. As colonizing blackflies in the laboratory is notoriously difficult, collecting field data to understand the kinetics of the antibody responses over a specified time frame would be helpful, particularly in settings with strong seasonality, in which vector biting ceases or greatly decreases for several months during the year. Regarding theoretical approaches, and as mentioned earlier, the exposure profiles predicted by age- and sex-structured onchocerciasis transmission models are broadly consistent with the proposed desensitization hypothesis. If this proves to be the case, we expect vector saliva-naïve children still to respond well to salivary antigens as an indication of continued exposure to vector bites pre- or post-MDA campaigns in areas with no vector control or without major ecological changes affecting vector density. Our assay could then be used to understand potential secular trends in vector biting rates due to anthropogenic change. Ideally, multiplex assays could be developed to test simultaneously for both exposure to vector bites and to parasite antigens.

## Conclusion

Serological anti-saliva assays are useful tools to complement information collected by HLCs by measuring human-vector contact and revealing heterogeneities in exposure at the individual and community level that cannot be unravelled by HLCs alone. By novel application to four onchocerciasis-endemic communities, this study successfully evaluated age- and sex-related demographic patterns in blackfly bite exposure. The analyses uncovered the possibility of age- and sex-specific immunotolerance or desensitization to blackfly saliva, likely resulting from cumulative blackfly exposure with age. Concomitant studies of infection levels in humans and flies, vector abundance, and immune responses to blackfly saliva and parasite antigens would greatly help to better understand transmission risk and intensity, and improve parameterization of transmission models with which to inform optimal interventions and surveillance strategies to achieve and protect onchocerciasis elimination.

## Supporting information

**S1 File. Supplementary Figures and Tables.** Fig A–Proportion of tested individuals. Fig B–Age distribution of the population shown per cluster and sex. Table C–Number of people sampled and tested for IgG and IgM per age group and sex. Fig D–IgG antibody distribution according to age shown per cluster. Fig E–IgM antibody distribution according to age shown per cluster. Table F–Summary of exponentiated regression coefficient estimates. Fig F–Correlation between IgG and IgM responses.
(DOCX)

## Acknowledgments

Special thanks go to Dr. Francis Veriegh, Frank Aboagye-Twum and Bright Idun for assisting in the blood sampling and the collection of blackfly salivary glands. We gratefully acknowledge the researchers at the Noguchi Memorial Institute for Medical Research in Accra for allowing us access to their laboratory facilities. We also thank all the participants of the study, who were crucial to the execution of the study.

## Author Contributions

**Conceptualization:** Mike Y. Osei-Atweneboana, Petr Volf, Maria-Gloria Basáñez, Orin Courtenay.

**Data curation:** Laura Willen.

**Formal analysis:** Laura Willen, Philip Milton.

**Funding acquisition:** Mike Y. Osei-Atweneboana, Petr Volf, Maria-Gloria Basáñez, Orin Courtenay.

**Investigation:** Laura Willen.

**Methodology:** Laura Willen, Philip Milton, Jonathan I. D. Hamley, Martin Walker, Petr Volf, Maria-Gloria Basáñez, Orin Courtenay.

**Project administration:** Petr Volf, Maria-Gloria Basáñez, Orin Courtenay.

**Resources:** Mike Y. Osei-Atweneboana, Petr Volf, Maria-Gloria Basáñez, Orin Courtenay.

**Supervision:** Jonathan I. D. Hamley, Martin Walker, Petr Volf, Maria-Gloria Basáñez, Orin Courtenay.

**Validation:** Laura Willen, Philip Milton.

**Visualization:** Laura Willen, Philip Milton.

**Writing – original draft:** Laura Willen.

**Writing – review & editing:** Laura Willen, Philip Milton, Jonathan I. D. Hamley, Martin Walker, Mike Y. Osei-Atweneboana, Petr Volf, Maria-Gloria Basáñez, Orin Courtenay.

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
