## [Decision Letter · Decision Letter 0]

2 Nov 2021

Dear Miss Willen,

Thank you very much for submitting your manuscript "Demographic patterns of human antibody levels to Simulium damnosum s.l. saliva in onchocerciasis-endemic areas: an indicator of exposure to vector bites" for consideration at PLOS Neglected Tropical Diseases. As with all papers reviewed by the journal, your manuscript was reviewed by members of the editorial board and by several independent reviewers. The reviewers appreciated the attention to an important topic. Based on the reviews, we are likely to accept this manuscript for publication, providing that you modify the manuscript according to the review recommendations. 

All reviewers acknowledged the novelty and quality of this manuscript. Please address the points made by the reviewers, especially expand the discussion on the limitations as mentioned by reviewer #2 and address the comments made by reviewer #3 on sample size calculation in dependence on age and the logistic regression model.

Sincerely,

Marc P Hübner, Ph.D.

Associate Editor

Sara Lustigman

Deputy Editor

All reviewers acknowledged the novelty and quality of this manuscript. Please address the points made by the reviewers, especially expand the discussion on the limitations as mentioned by reviewer #2 and address the comments made by reviewer #3 on sample size calculation in dependence on age and the logistic regression model.

Reviewer's Responses to Questions

**Key Review Criteria Required for Acceptance?**

**Methods**

-Are the objectives of the study clearly articulated with a clear testable hypothesis stated?

-Is the study design appropriate to address the stated objectives?

-Is the population clearly described and appropriate for the hypothesis being tested?

-Is the sample size sufficient to ensure adequate power to address the hypothesis being tested?

-Were correct statistical analysis used to support conclusions?

-Are there concerns about ethical or regulatory requirements being met?

Reviewer #1: Methods are clearly stated with appropriate statistical analyses supported by power calculations.

Reviewer #2: Methods are clear and well articulated. A couple of points 

Line 145: you say age categories from 5-10, upwards etc. but please amend the recruitment section above to clarify that children 4 and below were not recruited. 

Line 149: I am not sure quite what you mean by: with some variability in percentages tested in the older three age groups. Was this because there were fewer people in these groups?

Line 196: Thank you for the sample size calculations for the IgG, but how did you calculate this for the IgM. The only thing lacking is the explanation of the smaller sample size for the IgM.

Line 211: …..were tested, treating….. please include this comma.

Supplementary figure B1: why were fewer samples from older males used in the IgM ELISAs?

Reviewer #3: - the objectives of the study clearly articulated with a clear testable hypothesis stated

-the study design appropriate to address the stated objectives

-the population is clearly described and appropriate for the hypothesis being tested

-the sample size calculation needs some some improvement support conclusions

-No concerns about ethical or regulatory requirements

**Results**

-Does the analysis presented match the analysis plan?

-Are the results clearly and completely presented?

-Are the figures (Tables, Images) of sufficient quality for clarity?

Reviewer #1: Figures and tables are clearly presented and described. However, Table 3 can be deleted as it is redundant with respect to figure 5. In table 2, antibody titres are described as logged but this does not seem to be the case.

Reviewer #2: Results are clear, well 

Figures: please say 5-<10 years for all the graphs with the ages on, rather than <10 years

Please include IgG and IgM in the antibody responses axes labelling, to make it really clear which figure relates to which antibody. As you do for the predicted IgG responses for example.

Reviewer #3: This section may need improvement based on the previous comments

**Conclusions**

-Are the conclusions supported by the data presented?

-Are the limitations of analysis clearly described?

-Do the authors discuss how these data can be helpful to advance our understanding of the topic under study?

-Is public health relevance addressed?

Reviewer #1: The discussion is appropriate and is transparent about the limitations of the study. Could the authors comment on whether the antibody data could be combined with a spatial analysis in future? Notwithstanding the impact of occupation and travel on exposure, it would be expected that biting rates would decline quite rapidly with distance from blackfly breeding sites.

Reviewer #2: Yes the conclusions are supported, but the limitations should be discussed in greater detail and how this links with public health relevance, in that more work is needed for comparisons, before antibody responses can be used as a proxy for biting exposure.

Limitations:

Whilst you discuss that the antibody declines with age could be due to the two very different transmission potential reasons: namely either reducing exposure with age, or maintained high exposure and a reduced antibody response, you should therefore also talk about the limitations that this has on the usefulness of this method to help inform biting rates in models, unless more data are collected to directly link biting rates with saliva antibody responses. You allude to it in the conclusion, but not in the main discussion, so please amend this. you could also talk about the usefulness in children still to assess biting rates in saliva naïve individuals, such as the OV16 type monitoring but for transmission potential should MDA stop.

Another further limitation could be that the blackflies used to collect saliva were only from one location. Could this have affected the ELISA antibody assays at all? Is there any evidence that the species might be location specific for reactions at all? As you collected them from the highest transmission site and had the lower responses there maybe this might be another factor in this?

Reviewer #3: The conclusion will need to be improved based on the answers to the previous comments

**Editorial and Data Presentation Modifications?**

Reviewer #1: The manuscript is very well written, prepared and presented but the reference details should be checked for completeness. Refs. 42 and 47 by "Veriegh FBD" appear to be duplicates and the type of reference is unclear.

Reviewer #2: A few very minor changes other than in the methods, results and conclusions described above:

Abstract/summary

Line 48: identifying

Line 48: comma after Traditionally,

Line 54: and, rather than or

Introduction

First sentence needs splitting in to two.

Line 65: please put ‘stop-MDA surveys’ in inverted commas, as not everyone is as used to the term as us.

Line 75: ‘inefficient to’ replace with ‘unable to’

Line 77: please change to ‘…exposure to blackfly vector bites…’

Line 367: greater rate of decline: please be specific in what is declining here?

Reviewer #3: (No Response)

**Summary and General Comments**

Reviewer #1: This is an important study, the first of its kind, and the authors should be congratulated on presenting a clearly written and unfussy manuscript that puts across the key findings in a succinct and accessible manner. My comments above are very minor and I have no significant criticisms. However, the raw data have not been made available (or the accession no. is lacking) and no reason is provided for withholding it.

Reviewer #2: This is a nice paper with well presented results, but the limitations should be addressed more clearly, and also how the results indicate that more work is needed before the method can be used to help inform transmission models, as so nicely introduced at the start. But the science and writing are good, and with a more detailed discussion of the limitations of the study and the findings the paper will be a good addition to the literature and helps to address the ongoing issue of the HLCs.

Reviewer #3: PLOS Neglected Tropical Diseases ( PNTD-D-21-01444)

 Demographic patterns of human antibody levels to Simulium damnosum s.l. saliva in onchocerciasis-endemic areas: an indicator of exposure to vector bites

General comments

This study is a follow up of the work published by Willen et al. 2021, describing the human immune responses against salivary antigens of Simulium damnosum s.l., as new epidemiological marker for exposure to blackflies bite in onchocerciasis endemic areas. The present work used information from previous study and was carried out in four onchocerciasis endemic communities in Ghana. The study objective is to understand sex-or age-related demographic patterns in vector exposure. This study has demonstrated the possibility of sex-and age-specific immunotolerance or desensitization to blackly salivary following the likely cumulative exposure with age. The findings from this pioneer work are likely to contribute to more robust study design that could synchronize microfilarial infection intensity, vector abundance , infection rate and immune responses to blackfly saliva to better understand the transmission risk and intensity and to optimize vector control and surveillance. 

Methods:

(i)The sample size calculation was based on gender effect and not age effect. Meanwhile the data analysis is more dominated by the age effect, could that influence the findings?

(ii)Logistic regression analysis model could have been more robust to express the association between exposure factors (age, gender, residence, biting rate etc,) and immune responses to Simulium saliva in human. Why the authors did not choose the logistic regression model? I will suggest that the authors redo the data using logistic regression model

Specific Observations

L90-98: sentence is too long. 

L161: …….kept cool for 2–3 hours……. Describe the cooling system

L114: package of drink powder….. describe better the packaging system 

L229-229: “however both IgG and IgM median antibody responses declined with age…. Fig 3D does not seem to reflect that for IgM

L 229-230: A breakdown of the IgG and IgM antibody distribution with age per individual cluster is visualized in Figs D and E in S1 File. There is not figure captioned Fig D and E without a number, do you mean Figs 4 D and E?

L407: replace “cheap” with “ cheaper” and when you say relatively cheaper is compared to what?. 

L 407-414: The conclusion needs to be rephrased. say if the initial objective of the study have been reached.

PLOS authors have the option to publish the peer review history of their article (what does this mean?). If published, this will include your full peer review and any attached files.

Reviewer #1: No

Reviewer #2: No

Reviewer #3: No

Figure Files:

Data Requirements:

Reproducibility:

References

---

## [Editor Report · Decision Letter 1]

17 Dec 2021

Dear Miss Willen,

We are pleased to inform you that your manuscript 'Demographic patterns of human antibody levels to Simulium damnosum s.l. saliva in onchocerciasis-endemic areas: an indicator of exposure to vector bites' has been provisionally accepted for publication in PLOS Neglected Tropical Diseases.

Best regards,

Marc P Hübner, Ph.D.

Associate Editor

Sara Lustigman

Deputy Editor

Congrats to the authors. You addressed all points made by the reviewers well.

---

## [Editor Report · Acceptance letter]

7 Jan 2022

Dear Miss Willen,

We are delighted to inform you that your manuscript, "Demographic patterns of human antibody levels to </i>Simulium damnosum</i> s.l. saliva in onchocerciasis-endemic areas: an indicator of exposure to vector bites," has been formally accepted for publication in PLOS Neglected Tropical Diseases.

Best regards,

Shaden Kamhawi

co-Editor-in-Chief

Paul Brindley

co-Editor-in-Chief
